# Climate drives long-term change in Antarctic Silverfish along the western Antarctic Peninsula

Andrew D. Corso [1✉], Deborah K. Steinberg [1], Sharon E. Stammerjohn [2] & Eric J. Hilton[1]

Over the last half of the 20th century, the western Antarctic Peninsula has been one of the most rapidly warming regions on Earth, leading to substantial reductions in regional sea ice coverage. These changes are modulated by atmospheric forcing, including the Amundsen Sea Low (ASL) pressure system. We utilized a novel 25-year (1993–2017) time series to model the effects of environmental variability on larvae of a keystone species, the Antarctic Silverfish (*Pleuragramma antarctica*). Antarctic Silverfish use sea ice as spawning habitat and are important prey for penguins and other predators. We show that warmer sea surface temperature and decreased sea ice are associated with reduced larval abundance. Variability in the ASL modulates both sea surface temperature and sea ice; a strong ASL is associated with reduced larvae. These findings support a narrow sea ice and temperature tolerance for adult and larval fish. Further regional warming predicted to occur during the 21st century could displace populations of Antarctic Silverfish, altering this pelagic ecosystem.

[1] Virginia Institute of Marine Science, William & Mary, Gloucester Point, VA, USA. [2] Institute of Arctic and Alpine Research, University of Colorado, Boulder, CO 80309, USA. ✉email: adcorso@vims.edu

The Antarctic Silverfish (*Pleuragramma antarctica*; Nototheniioidei) can comprise over 90% of adult and larval fish biomass in coastal areas of the Southern Ocean[1,2], and is the only endemic Southern Ocean fish that maintains an entirely pelagic life history[3]. Due to their abundance and availability within the upper water column, Antarctic Silverfish are important prey for higher predators, such as seals, penguins, other seabirds, and other fishes[4,5]. Antarctic Silverfish have a circumpolar distribution with populations having genetic connectivity along the continental shelf[6]. However, the absence of an Antarctic Slope Front Current in the western Antarctic Peninsula (WAP) region (Fig. 1) isolates this Antarctic Silverfish population from other established areas of reproduction in the species[6].

The central role of Antarctic Silverfish in the WAP food web has led to concern over the health of this isolated population due to regional climate change. Physiological adaptations that allow endemic fishes (most notably, members of Notothenioidei) to survive in the cold waters surrounding Antarctica also make them susceptible to ocean warming. Manipulation experiments show that an increase in water temperature of 5 °C can result in mortality[7,8] and reduced food assimilation rate[9] in some notothenioid fishes. From 1945 to 2009, the mean ocean temperature at mid-depths (100–300 m) along the WAP increased by 1 °C; these mid-depth ocean waters are already 4 °C warmer than the seawater freezing temperature[10]. Concurrently, mean annual and winter air temperature increased by 3 and 6 °C, respectively[11,12]. Although this rapid warming likely caused no direct mortality in adult Antarctic Silverfish, it may have led to detrimental physiological responses at all life stages, especially in the less mobile larval stages[13].

Regional warming in the WAP, combined with changes in winds, has also led to a decrease in sea ice, which plays a key and

**Fig. 1 Impact of the Amundsen Sea Low (ASL) on the West Antarctic environment. a** Averaged environmental conditions for the duration of an especially weak (shallow; relative central pressure, RCP, of −6) ASL event during March–April–May (MAM) 1993. The ASL central location is marked by the black asterisk and the Palmer Antarctica Long-Term Ecological Research program study region and sampling stations are depicted by eight lines of small black dots. Sea ice concentration anomalies are color shaded and the MAM 1993 mean ice edge contour (solid black line) and long-term (1979–2019) mean contour (dotted black line) are also marked. Subsampled wind anomalies are shown in vector format. The negative sea-level pressure (SLP) anomalies are shown for the most negative feature during MAM 1993 (dashed concentric contours). **b** The same averaged environmental conditions and symbology as in **a** but for the duration of an especially strong (deep; RCP of −16) ASL event during MAM 1996. The wind-vector legend for both 1993 and 1996 is boxed in the lower right corner.

unique role in Antarctic Silverfish life history. These fish deposit their eggs within sea ice, which also serves as a nursery area for newly hatched larvae[3,14,15]. The total elapsed days between sea ice advance and retreat, or annual sea ice duration, is used to monitor trends in the cryosphere. Between 1979 and 2018, annual sea ice duration along the WAP decreased by about 2 months[12]. Therefore, recent declines in sea ice and thus spawning habitat have been implicated in diminished Antarctic Silverfish abundance in the WAP region[13,16–19] and other coastal areas of the Southern Ocean[20–22]. However, abundance data of sufficient temporal and spatial scale for Antarctic Silverfish was previously unavailable to test this hypothesis.

Three atmospheric circulation patterns predominantly influence climate change in the WAP region – the Southern Annular Mode (SAM), El Niño Southern Oscillation (ENSO), and Amundsen Sea Low (ASL). Interconnections among these are complex, seasonally variable, and tied to anthropogenic impacts such as global greenhouse gas emissions[23]. The ASL, a climatological low-pressure center located in the Amundsen Sea (Fig. 1), has only recently been identified as the main factor behind WAP ocean warming[24–26], sea ice loss[27–29], and glacial retreat[30,31] over the last century.

The strength and location of the ASL influence meridional winds east and west of the ASL center[25] (Fig. 1a, b); stronger (i.e., deeper or spun-up) ASL events in the Amundsen Sea, for example, increase warm northerly airflow over the WAP region (i.e., along the eastern limb of the clockwise rotating low-pressure center), which reduces sea ice extent and concentration (Fig. 1b) and increases the temperature at the surface along the WAP[27]. In addition, the strength and location of the ASL influence zonal winds to the north and south of the ASL center. For example, an ASL that is centered southward (i.e., poleward) in the Amundsen Sea increases westerly wind anomalies near the continental shelf break[26], enhancing the flow of warm Circumpolar Deep Water (CDW) southward onto the shelf[30–32]. Although less is known regarding the impact of the ASL on CDW intrusions in the Bellingshausen Sea, a similar relationship is probable[33–36]. While several studies examined the effects of SAM and ENSO oscillations on Antarctic phytoplankton and zooplankton[37–39], no previous analyses focus on the potential impacts of the ASL strength or location on any organism.

Here we investigate connections between Antarctic Silverfish and the WAP environment across more than two decades. Specifically, we evaluate the impact of ocean temperature, sea ice dynamics, atmospheric circulation patterns, chlorophyll, and salinity on larval Antarctic Silverfish abundance. We use results from this analysis to offer predictions of regional pelagic food web response to future climate change.

## Results and discussion

**Ocean temperature and larval abundance.** We sorted, identified, and enumerated Antarctic Silverfish larvae ($n = 7086$) collected in a 25-year time series (1993–2017; Fig. 2a) of plankton net tows (see "Methods") as part of the Palmer Antarctica Long-Term Ecological Research (Palmer LTER) Program (Supplementary Fig. 1) and archived in the Nunnally Ichthyology Collection at the Virginia Institute of Marine Science. Zero-inflated generalized linear mixed-effects model (GLMM) predictions show the abundance of larvae during this period is closely tied to sea surface temperature, with higher abundance in colder water ($p < 0.001$; Fig. 2b and Supplementary Table 1). Approximately 45% of Antarctic Silverfish larvae were collected at sea surface

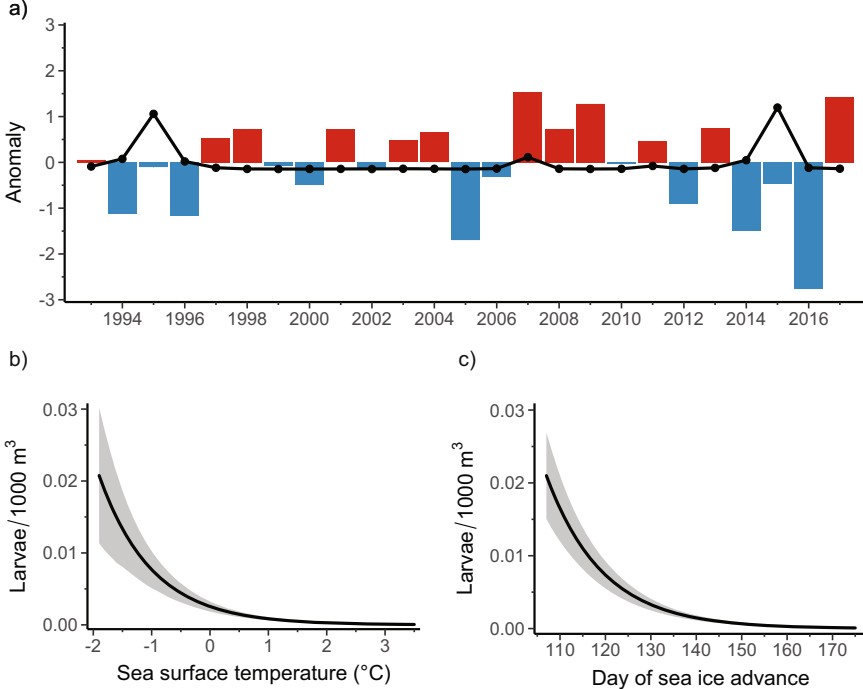

**Fig. 2 Relationships between sea surface temperature, sea ice, and Antarctic Silverfish abundance. a** Positive (red; warmer temperatures) and negative (blue; cooler temperatures) anomalies in the standardized mean sea surface temperature (see "Methods") for the Palmer LTER study region (Supplementary Fig. 1) during austral summer (December, January, February). Standardized anomalies in mean annual larval Antarctic Silverfish abundance (larvae/1000 m³) that were captured during January and February are overlaid (black dotted line). **b, c** Predicted impact (solid black lines) on larval Antarctic Silverfish abundance from **b** sea surface temperature ($p < 0.001$) and **c** lagged day of sea ice advance ($p < 0.001$) from the model (see "Methods"). Sea ice advance was temporally lagged in the model to align with life history patterns in adult Antarctic Silverfish abundance. The shaded regions represent the 95% prediction interval, which considers uncertainty from the fixed effects, zero-inflation, and random effects components of the final model (Supplementary Fig. 2a[112], see "Methods").

temperatures ranging from −2 to 0 °C, and 95% of larvae were captured in waters colder than 1.5 °C. Based on estimated marginal means, we predict that no larvae occur at sea surface temperatures of ≥1.7 °C (see "Methods").

Based on the time series, at least two consecutive years of anomalously cold surface temperatures are necessary to produce peaks in larval abundance (Fig. 2a). Palmer LTER net tows are deployed to a depth of 120 meters, encompassing the majority of habitat occupied by Antarctic Silverfish larvae[22,40]. Using data derived from CTD casts paired with net tows, we found Antarctic Silverfish abundance was significantly linked to colder temperatures averaged over the 120 m water column. However, sea surface temperature yielded optimal model performance (see "Methods").

**Impact of sea ice on spawning**. We next isolated the effects of sea ice dynamics and climatic variability on Antarctic Silverfish adults by lagging variables in the model[38,41]. Model results indicate the timing of sea ice advance during austral autumn (March–May) controls larval abundance in the following year ($p < 0.001$; Fig. 2c and Supplementary Table 1). Model performance (see "Methods") was considerably reduced when sea ice advance was lagged 2 years, or when there was no lag (Supplementary Table 2). To our knowledge, this is the first statistically significant relationship reported between sea ice and the long-term abundance of any Antarctic fish species.

There are relatively sparse data on the reproductive biology of Antarctic Silverfish, especially in the WAP region[17]. Adult fish likely spawn during late austral winter to early spring (July–September), eggs develop for approximately 4 months, and hatch in November and December (Fig. 3a). Larvae are then sampled annually by the Palmer LTER during January and February. However, there is evidence from the Ross Sea that

Antarctic Silverfish are skip spawners[42]. In austral autumn, adult fish migrate from their offshore pelagic habitat to coastal areas. Adults then improve their nutritional condition for a year before spawning the next season (Fig. 3a)[17,42].

We suggest that adult Antarctic Silverfish select their spawning area along the WAP based in part on the presence of sufficient sea ice cover during austral autumn (Figs. 2a and 3b, c). An early sea ice advance in late April or early May (days 110 to 130 in Fig. 2c) acts as a positive cue for migrating adults and increases spawning habitat (Fig. 3b). If advance is delayed, spawning habitat is reduced (Fig. 3c) and could cause adults to travel elsewhere or continue to postpone spawning. We therefore predict that sea ice advance beginning approximately on day 157 or earlier is necessary for spawning to successfully occur in this region (see "Methods"). Seasonal sea ice variability during the past 20 million years in the Southern Ocean has led to a high level of life-history plasticity among Antarctic Silverfish[3,16]. However, given the sea ice dependence indicated by our model, it is likely that the acute reduction in sea ice during the 20th century has significantly decreased spawning in the northern WAP. Consequently, there has been a lower abundance of mature adults observed in the northern WAP for several decades[17,18,43].

**Connections with ASL strength and location**. Larval Antarctic Silverfish abundance was also significantly correlated with the ASL relative central pressure (RCP), an index of ASL strength, ($p = 0.006$; Fig. 4a and Supplementary Table 1)[26], and the latitudinal location of the ASL ($p = 0.005$; Fig. 4b and Supplementary Table 1). We considered ASL RCP and latitude averaged over summer (DJF), autumn (MAM), winter (JJA), and spring (SON) months. Longitudinal location of the ASL was not included in this analysis because it exhibits a strong negative correlation with RCP during most of the year[44]. Lagged ASL strength and latitude

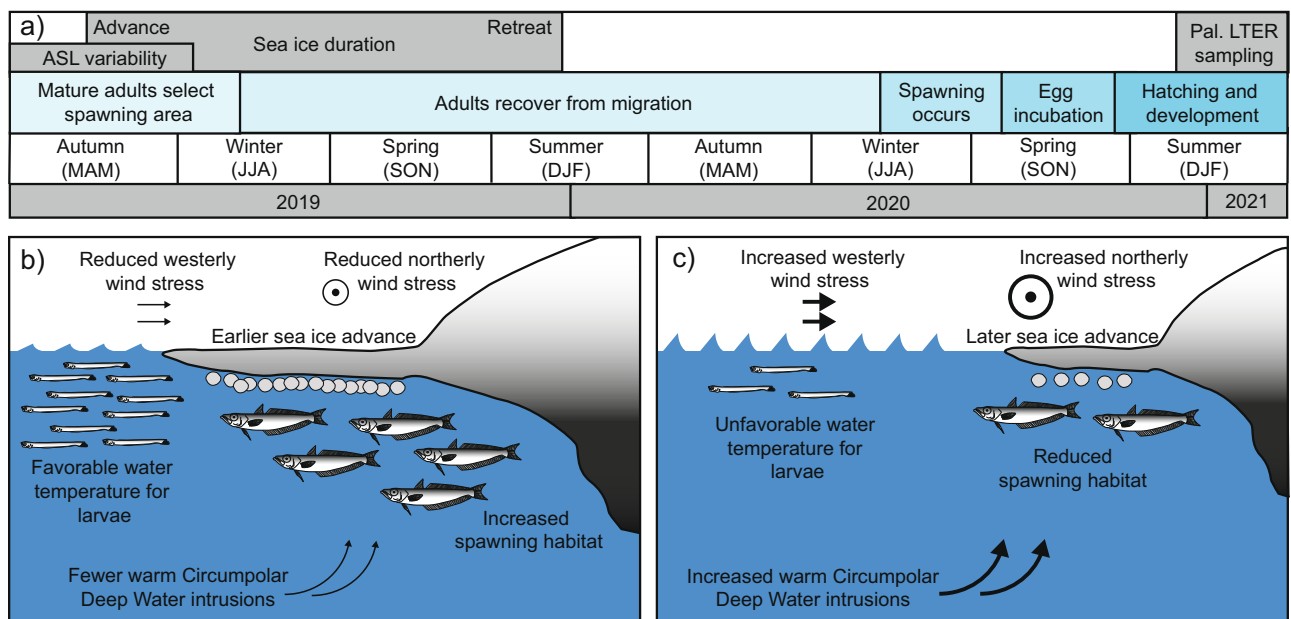

**Fig. 3 Timeline with optimal and suboptimal conditions for Antarctic Silverfish reproduction near the western Antarctic Peninsula (WAP). a** A timeline of Antarctic Silverfish skip-spawning behavior and proposed relationships with Amundsen Sea Low (ASL) variability and sea ice advance. **b, c** Schematic of **b** optimal and **c** suboptimal environmental conditions for larval and spawning adult Antarctic Silverfish near the WAP. Northerly and westerly wind stresses are both modulated by the ASL strength (i.e., RCP) and location (i.e., latitude). Intrusions of Circumpolar Deep Water (CDW) are associated with anomalous westerly winds; sea ice advance is influenced by a combination of wind stress, CDW, precipitation, and other factors; and near-surface water temperatures are determined by atmospheric heat, CDW intrusions, ice melt, and other factors.

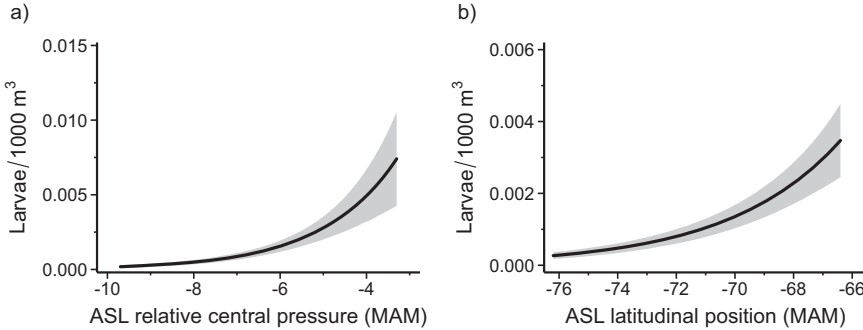

**Fig. 4 Predicted impact of Amundsen Sea Low relative central pressure and latitudinal location on larval Antarctic Silverfish abundance. a, b** Predicted impact (solid black lines) on larval Antarctic Silverfish abundance from **a** the relative central pressure (RCP) of the Amundsen Sea Low (ASL) during austral autumn (March–April–May [MAM]; $p = 0.006$) and **b** the latitudinal location of the ASL during austral autumn ($p = 0.004$) from the model (see "Methods"). Smaller RCP values correspond with stronger (i.e., deeper) ASL events and smaller latitudinal values correspond with southward (i.e., poleward) locations of the ASL. The ASL RCPs and locations were temporally lagged in the model to align with life history patterns in adult Antarctic Silverfish abundance. The shaded regions represent the 95% prediction interval, which considers uncertainty from the fixed effects, zero-inflation, and random effects components of the final model (Supplementary Fig. 2a[112], see "Methods"). It is important to note only one value of ASL RCP was greater (less negative) than −4.5, which accounts for the increased uncertainty at less negative values (>−4.5). Additionally, we observed no significant collinearity between ASL RCP and latitude in this analysis (see "Methods").

averaged over autumn produced optimal model AIC (see "Methods"). These relationships further support our hypothesis that adult Antarctic Silverfish are selecting their spawning habitat during autumn when sea ice is also beginning to advance (Fig. 3a). Autumn seasons with stronger (more negative) RCPs (Fig. 4a) and a poleward ASL location (Fig. 4b) were associated with diminished spawning success reflected by lower abundance of the larvae sampled in the following year. There was no correlation between larval abundance and lagged ENSO indices during any season. We observed a non-significant negative relationship between lagged Marshall SAM index and larval abundance during autumn ($p = 0.10$) and summer ($p = 0.09$); larval abundance was higher following years with a more negative SAM. However, including the SAM index for any season did not improve model performance (Supplementary Table 3) and thus was not included in the final model (see "Methods"; Supplementary Fig. 2a).

The positive relationship between the lagged latitude of the ASL and larval abundance is suggestive of CDW intrusions impacting adult fish. Increased intrusions of CDW onto the Amundsen continental shelf that occur during a poleward shift in the ASL (Fig. 3c) are well below the thermocline[32], with the CDW temperature maximum around 400 m[45]. This warm layer of water, with temperatures approaching 2 °C[46], expands as more CDW is transported onto the shelf[32]. This water mass is well above the optimal thermal environment for Antarctic Silverfish, particularly as the physiological costs of spawning further impact their limited thermal tolerance[47]. Adult Antarctic Silverfish, which occupy a depth range of 0 to 900 m[3], might therefore delay spawning or attempt to relocate when oceanic and sea ice conditions are unfavorable in the Bellingshausen Sea due to ASL strength and location (Fig. 3c). However, additional ocean modeling is required to verify specific causes of the correlation between ASL strength, location, and adult fish.

Although there are connections between the West Antarctic cryosphere inclusive of the WAP and ASL[27,28], there were no indications of collinearity between WAP ice advance, sea surface temperature, autumn ASL RCP, and the ASL latitudinal location in this analysis (see "Methods"). The positive correlation between ASL RCP and larval abundance is likely due to the warm northerly winds associated with strong (deep) RCPs affecting sea ice in pathways that are not captured by the ice advance parameter.

One such pathway is that strong ASL events reduce the areal extent of sea ice by advecting and compacting the ice edge southward in the Bellingshausen Sea[48,49], which also reduces frazil ice production, the overall impact being less ideal spawning habitat for Antarctic Silverfish. In more southerly locations (e.g., the coastal Ross Sea), Antarctic Silverfish eggs are usually associated with the presence of a sub-ice platelet layer (SIPL)[14]. The SIPL is highly porous, which facilitates nutrient exchange with seawater and supports exceptionally high algal biomass[50,51]. This productive habitat is an ideal nursery area for larval Antarctic Silverfish. However, frequent intrusions of warm CDW in the Bellingshausen Sea likely prevent SIPLs from forming[50], and platelet crystals have been observed only once in this region[52]. Instead, Antarctic Silverfish in the WAP area likely search for similarly highly porous and productive habitat, such as dense aggregations of frazil ice[53,54].

Furthermore, sea surface temperatures (<20 m) increased in the WAP region by approximately 2 °C during the 20th century (1955–1998)[55]. Although this abrupt warming has recently slowed[56], it is expected to intensify in the coming decades[32,57]. The altered near-surface winds associated with prolonged deepening and a poleward shift in the ASL, modulated by the SAM[58] and Pacific variability[59], contribute to past and future surface warming in the WAP region[26]. Therefore, we posit that long-term strengthening of the ASL creates unfavorable water temperatures for Antarctic Silverfish larvae in the WAP region (Fig. 3c)[32,57].

**Impact of chlorophyll and salinity on larval abundance.** In addition to sea surface temperature, larval Antarctic Silverfish abundance was positively correlated with chlorophyll concentration ($p < 0.001$; Supplementary Fig. 3a and Supplementary Table 1), indicating bottom-up control in this food web. Phytoplankton is grazed by the early life stages of copepods (i.e., copepodites) which in turn are the primary prey of larval Antarctic Silverfish[3]. While the focus of the present study is on modeling environmental variables, confounding influences of prey field dynamics due to a changing climate are also possible. The phytoplankton community shifts during warmer years with low sea ice[12,60], potentially altering copepod abundance and composition[61]. In addition to the physiological cost of warmer waters on larval Antarctic Silverfish, climate change could cause

their hatching period to become out of sync with peak abundances of their prey[13,22,62].

A higher abundance of larvae also occurred in areas of lower surface salinity ($p < 0.001$; Supplementary Fig. 3b and Supplementary Table 1). Surface freshwater inputs from melting sea ice and glaciers lead to a more stratified surface layer and increased chlorophyll;[63] such a stratified environment with ample food could be preferred by Antarctic Silverfish larvae. The relationship between salinity and larval abundance also points to the importance of sufficient sea ice cover. High-salinity brine is exported downwards in the water column during autumn sea ice growth and advance[63,64]. Consequently, surface salinity is fresher following winters with high sea ice extent[64].

Slósarczyk[65] found Antarctic Silverfish larvae were more abundant in areas of high salinity in the WAP region (34.1–34.6)[13]. However, their analysis used a single year of pelagic net tows, and the mean standard length (SL) of post-larval and juvenile Antarctic Silverfish they collected (75 mm) was over five times the mean SL of larvae used in our study (11.9 mm; Supplementary Fig. 4). Osmoregulation functions change in fishes during their development[66]; therefore, post-larval Antarctic Silverfish could possess a greater ability to tolerate more saline water[13]. No experiments to date address the salinity thresholds of Antarctic Silverfish during any stage in their life history.

**Outlook of WAP climate and Antarctic Silverfish.** Our results support the hypothesis that sea ice and ocean temperatures, modulated by the ASL, impact Antarctic Silverfish spawning behavior, resulting in changes in the abundance of their larvae along the WAP. Two anthropogenic forces jointly influence the prevailing circumpolar westerly winds and the strengthening/weakening of the ASL: the ozone hole located over Antarctica and greenhouse gas emissions; both tend to strengthen the westerly winds poleward and deepen the ASL[26,67,68]. However, the Antarctic climate is characterized by high regional and seasonal variability, both natural and forced, making it challenging to identify attribution[27]. For example, a recent cooling along the WAP and increases in sea ice extent were attributed to Pacific decadal variability[11], though other factors were likely also at play. Nonetheless, it is expected that the recent reversal in WAP climate trends will not persist with continued increases in greenhouse gas emissions. The Antarctic ozone hole is predicted to recover entirely by ~2100, while most scenarios predict increasing greenhouse gas emissions through 2100[69]. High carbon dioxide emission scenarios will likely result in a prolonged trend of strong ASL events and atmospheric warming in the WAP region despite ozone recovery[57,70,71]. Antarctic Silverfish have encountered periods of cyclic warming during their evolutionary history[13]. However, with precipitous climate warming due to continuing greenhouse gas emissions, Antarctic Silverfish could disappear from the WAP region entirely, triggering changes in other components of the pelagic food web.

**Possible ecosystem responses to Antarctic Silverfish decline.** For the last 6,000 years, Antarctic Silverfish have dominated Adélie penguin diets in the WAP region during periods of especially cold temperatures[72]. Furthermore, the substantial decline of Adélie penguins in the northern WAP[73] is coincident with a long-term decrease in Antarctic Silverfish in their diet[43]. While there are several causal mechanisms driving population changes in Adélie penguins[74,75], these changes are possibly exacerbated by diminished Antarctic Silverfish abundance resulting from recent warming and reduced sea ice[74]. Antarctic Silverfish, especially spawning adults, are high-energy prey items[43,76]. Penguins are known to switch to entirely invertebrate-based diets

(e.g., krill or squid) during warm periods[43]. However, the loss of lipid-rich Antarctic Silverfish in Adélie penguin chick diets induces low fledging weights, a vital determinant of future recruitment[43,75].

Antarctic Silverfish are also a primary prey item for the commercially important Antarctic Toothfish (*Dissostichus mawsoni*) in the Ross Sea[4,77–80] and likely in other coastal Antarctic regions including the Bellingshausen and Amundsen Seas. There are many remaining gaps in knowledge regarding the life history of the Antarctic Toothfish. However, Antarctic Silverfish are likely to be an especially important component of their diet as they prepare for the migratory phase of their life history[77,81]. Shifts in the migratory behavior of Antarctic Silverfish due to changes in environmental conditions could have bottom-up effects on the condition and location of regional populations of Antarctic Toothfish[82]. Furthermore, some species of seals, especially the Weddell seal (*Leptonychotes weddellii*), heavily consume both Antarctic Toothfish and Silverfish[83–85]. Changes in the occurrence of Antarctic Silverfish could cause Weddell seal populations to increase their predation pressure on Antarctic Toothfish, or other fishes.

It is challenging to predict specific top-down effects of a severe reduction in Antarctic Silverfish on the regional zooplankton community as the dietary preferences of Antarctic Silverfish broaden across ontogeny[3], with adults maintaining dietary flexibility by consuming a wide size range of copepods, pteropods, euphausiids, amphipods, other zooplankton, and early life stages of notothenioid fishes[3,86]. Some abundant copepod species, particularly *Metridia gerlachei*, could experience a positive shift in abundance due to reduced predation from Antarctic Silverfish[40,87]. With one of the highest grazing rates of Antarctic copepods[88], an increase in *M. gerlachei* abundance could significantly alter microbial food web composition[88,89].

Trophic relationships and life history of Antarctic Silverfish are mostly known from the Ross and Weddell seas, thus it will be critical to further characterize the role of this keystone fish in the vulnerable WAP pelagic ecosystem. This study also demonstrates the value and importance of both long-term sampling programs[90,91] and natural history collections[92]. Curated time series of larval fishes are rare but invaluable resources to determine the causes of adult population changes[93,94]. Finally, the ecosystem-level consequences of climate change must be considered in the context of air–sea interactions, such as the ASL, to predict food web shifts more accurately and manage natural resources in the region.

## Methods

**Sampling.** Larval fishes used in this study were collected in accordance with the protocols of the Palmer LTER program[38] and were obtained as preserved specimens cataloged in the Virginia Institute of Marine Science (VIMS) Nunnally Ichthyology Collection. Scientists on Palmer LTER cruises collect multidisciplinary data in a fixed-sampling grid in the Bellingshausen Sea along the WAP (Supplementary Fig. 1)[95,96]. Zooplankton and larval fishes are sampled annually (for this study, years included 1993–2017) during austral summer (January–February) using a 2-m$^2$ frame Metro net (700-μm mesh) towed to ~120 m depth. A General Oceanics flowmeter positioned in the center of the net mouth was used to calculate each tow volume. All organisms in the tows are identified to family-level, measured in volume, and preserved in a formaldehyde solution.

From 1993 to 2007, tows were conducted from the 600 grid line to the 200 grid line[38]. In 2008, the "far south" lines (100, 000, and −100) were added to the Palmer LTER grid (Supplementary Fig. 1). The far south stations were included in this analysis and temporal autocorrelation was accounted for (see Model development). We included additional tows that were conducted from 1993 to 2017 at Process Study stations located between grid lines. We ran the final model without these Process Study and far south stations and the relationships were unchanged.

As larvae in this study are collected during the same fixed two-month period annually, it is important to note Antarctic Silverfish could be altering the timing of spawning during reduced sea ice conditions, resulting in modified hatching and subsequent peaks in larval abundance[16]. However, there has been a minimal

deviation in the standard length of Antarctic Silverfish larvae during the Palmer LTER (Supplementary Fig. 4) and no yolk-sac larvae have been sampled.

**Identification**. Antarctic Silverfish is among the most easily identified of the known larval fishes endemic to the Southern Ocean due to possession of two distinct dorsal pigment rows and a lack of abdominal pigmentation. All larvae in this study were identified based on Kellermann[97]. The majority (>70%) were identified to species level by Andrew D. Corso, and the remaining fishes were identified by Dr. Peter Konstantinidis (Oregon State University) and other scientists associated with the VIMS Nunnally Ichthyology Collection.

**Statistics and reproducibility**. The density (abundance divided by tow volume [in units of 1000 m$^3$]) of Antarctic Silverfish for each tow from 1993–2017 was modeled against environmental variables. We considered ocean temperatures and salinity from near the sea surface (5 m depth), the bottom of the tow depth (120 m), and averaged from 120 m to the surface. All temperature and salinity measurements were collected using CTD casts that were spatially and temporally paired with net tows. Discrete measurements of Chlorophyll a concentration measured in water collected in CTD casts were integrated to 100 m and also paired spatially and temporally with net tows. Bathymetry and time of day during net tows were also evaluated. Sea ice variables considered were derived from satellite imagery (Scanning Multichannel Microwave Radiometer and Special Sensor Microwave/Imager (SMMR-SSM/I))[98] and included duration, extent, day of the retreat, and day of advance.

Annual indices of climatic teleconnections (e.g., ENSO, SAM, and ASL) were included in the model. The ENSO index is based on sea surface temperatures (referred to as the multivariate ENSO index [MEI]; https://psl.noaa.gov/enso/mei/) and the SAM (http://www.antarctica.ac.uk/met/gjma/sam.html) index is based on sea level pressure. These climate indices are seasonally adjusted[99]. The RCP and latitudinal location of the ASL were obtained from Hosking et al.[100]. The RCPs from all four seasons (DJF, MAM, JJA, and SON) were evaluated. Multicollinearity between the conditional model parameters was tested with the Variance Inflation Factor (VIF) and correlation coefficient[101]. All parameters in the final model had VIF values of <2, well under cutoff range for moderate correlation (5–10)[102].

Zero-inflated GLMMs following a Tweedie distribution[103] were developed in R[104] and used to model the relationships between environmental variables and larval abundances. Antarctic larval fishes primarily exist in the upper 300-m of the water column[22] and are incidentally captured by Palmer LTER trawls targeting zooplankton (e.g., krill, salps, copepods). Resultingly, there is a significant overrepresentation of "zero fishes captured" each year at stations along the Palmer LTER sampling grid. Zero-inflated GLMMs were selected over zero-altered, as any zeros in the larval Antarctic Silverfish time series are likely "false," resulting from an imperfect sampling design[105].

Using an autocorrelation function (ACF) and partial autocorrelation function (PACF), temporal autocorrelation was observed, likely due to the fixed-sampling grid design[106]. An autoregressive covariance (AR1) is used in the GLMMs to accommodate the temporal autocorrelation between contiguous years[107]. There are no indications of spatial autocorrelation in the time series, although "nuggets" detected using variograms suggest fine-scale spatial variation and measurement noise[108]. Palmer LTER sampling stations (i.e., net tow coordinates) were treated as a random effect to account for unobserved spatial heterogeneity[109]. Generalized additive mixed-effects models (GAMMs) were also considered as several environmental variables exhibit marginal non-linearity. However, the lack of combined zero-inflation and autocorrelation structures for GAMMs limited their performance compared to GLMMs that account for these complexities[110]. The final model (Supplementary Fig. 2a; Supplementary Table 1) was selected based on hypothesis testing, parsimony, and minimizing the Akaike information criterion (AIC)[111]. Diagnostic residual plots (Supplementary Fig. 5) were also used to evaluate the model performance.

Predicted values (or estimated marginal means) were developed using the ggeffects package[112] in R[104]. The predicted values of larval density were conditioned on the fixed effects, zero-inflation, and random effects components of the final model (i.e., "re.zi")[112]. The 95% prediction intervals surrounding the predictions consider the mean random effect variance and are generally larger than confidence intervals due to this additional level of uncertainty[112–114]. Predictions of environmental cutoff values where no larvae occur were determined when the upper 95% prediction interval included 0.000 larvae/1000 m$^3$.

**Figure development**. Figure 1. Monthly sea ice concentration anomalies are derived from the SMMR/SSM/I satellite time series based on the Goddard Space Flight Center (GSFC) Bootstrap algorithm[115,116]. The sea ice concentration data are gridded to 25 km and are provided by the EOS Distributed Active Archive Center (DAAC) at the National Snow and Ice Data Center (NSIDC, University of Colorado at Boulder, http://nsidc.org). The numerically analyzed monthly 10-m height winds and sea-level pressure anomalies are from the fifth-generation European Centre for Medium Range Weather Forecasts (ECMWF) Reanalysis (ERA-5)[117] and are provided by the Climate Data Store (CDS, https://cds.climate.copernicus.eu/).

Figure 2. Sea surface temperatures for the Palmer LTER study region were determined by the NOAA optimal interpolation (OI) sea surface temperature analysis (version Reyn_SmithOI.v2) using in situ and satellite sea surface temperatures[118], plus sea surface temperatures simulated by sea ice cover. Annual anomalies of larval density and sea surface temperature were standardized by subtracting the sample mean then dividing by the sample standard deviation.

**Reporting summary**. Further information on research design is available in the Nature Research Reporting Summary linked to this article.

## Data availability
All data analyzed in this study are publicly available. The Antarctic Silverfish larvae are archived in the VIMS Nunnally Ichthyology Collection and data are publicly available on the VIMS Specify web portal (https://www.vims.edu/research/facilities/fishcollection/search_collection/index.php). The associated Palmer LTER environmental variables are available online from the Palmer LTER web portal. See https://pallter.marine.rutgers.edu for more information.

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

## Acknowledgements
We would like to thank Joseph Cope (VIMS) for assisting in the collection of fishes and data analysis. We also thank Sarah Huber (VIMS) and Peter Konstantinidis (Oregon State University) for their contributions to the identification of fishes and curatorial support; Adena Schonfeld (VIMS) and Robert Latour (VIMS) for their assistance in statistical analysis; Carlos Moffat (University of Delaware), Scott Hosking (British Antarctic Survey), William Fraser (Polar Oceans Research Group), and Christopher Jones (NOAA SWFC) for their aid in the interpretation of results. We are grateful for the multiple contributions of Marino Vacchi, Laura Ghigliotti, and their colleagues (Institute of Marine Sciences – National Research Council) regarding the life history of the Antarctic Silverfish. We thank the captains, crew, and support staff of the R/V *Polar Duke* and ASRV *Laurence M. Gould*, the support of personnel at Palmer Station, Antarctica, and the Leidos Antarctic Support Contractors. This work was funded by the National Science Foundation Antarctic Organisms and Ecosystems Program (PLR-1440435 and OPP-2026045 for specimen and environmental data collection) and Division of Biological Infrastructure (DBI-1349327 for specimen preservation and analysis), and the VIMS John Olney Fellowship. This is contribution 4067 of the Virginia Institute of Marine Science, William & Mary.

## Author contributions
D.K.S. led the collection of fishes; E.J.H. oversaw the preservation and identification of fishes; all authors conceived the ideas; A.D.C. and S.E.S. analyzed the data; all authors interpreted the results. A.D.C. wrote the manuscript; all authors edited the manuscript.

## Competing interests
The authors declare no competing interests.
