## [Peer Review File · Communications Biology]

Reviewers' comments:

Reviewer #1 (Remarks to the Author):

"Climate drives long-term change in Antarctic Silverfish along the western Antarctic Peninsula" reports on a 25-year dataset of larval abundance of silverfish on the western Antarctic Peninsula (WAP) and connects abundances to atmospheric forcing of the Amundsen Sea Low (ASL). The novelty of the study is that making a connection between ASL and abundance of any vertebrate has not been done before, and recent research suggests exciting connections between the ASL and climate impacts, especially on the WAP. That ASL fluctuations might explain abundance of larval silverfish in the region over 25 years is interesting.

I thought this was a well-written paper, certainly thought provoking, but feel the manuscript could be improved to better emphasize the findings, particularly for a non-expert in the realm of atmospheric forcing (which I am not) or Southern Ocean ecology (which I am familiar with). Here I provide general feedback, with specific questions below. The overarching takeaway for me throughout was that there was a lack of central focus - for example, laying out specifically your hypotheses and/or objectives in the introduction would help. I went back and forth in the manuscript, reminding myself what the questions were, and I think if they had been laid out clearly - and referred to consistently in the remaining sections - that would be very helpful to really drive your point home. For example in the Results section, there is quite a bit of Discussion, in my view, which distracted me from your key findings. It was difficult to connect the results of your statistical models with your interpretation. I think a straight-forward presentation and keeping the sections separate and distinct from each other would have helped me comprehend the science a bit better.

I was also surprised to see a lack of discussion or focus on the implications of results for marine predators especially given the rebuttal letter to the editor, which noted the importance of penguin dynamics to Nature Climate Change readers as justification to reconsider this manuscript... I would agree with your logic, but then I would recommend making the connection to this point stronger in your manuscript. The fact that the ASL affects sea ice dynamics and wind, that that in turn affects abundance/distribution of larval silverfish is in itself interesting, but I think to round out the manuscript, paying more attention to this point you made (i.e., their importance in the food web as both predators and prey) would strengthen this justification for this manuscript to be considered for publication, as per your rebuttal letter. I would also like to note that silverfish are a key component in Weddell seal diet (far less so for crabeater seal, as in your rebuttal letter).

Here are some specific questions and feedback:

Line 52: Here it might be worthwhile to note that in particular they can deposit eggs in frazil ice, which if I am not mistaken can be formed via strong, cold katabatic winds. If katabatic winds are less frequent, in addition to ice extent and ocean temperature (super-cooled waters for frazil ice formation), that could very well affect eventual larval abundance? Might be worth mentioning if this is correct, to strengthen your point.

Line 89: sampled? or detected? or both?

Line 106-108 is Discussion, I think.

Line 110-116 is probably better suited for Discussion. Typically, if something needs a reference to another study, you are inherently not talking about the results of your own study and in my view, it is not suitable for the results section - unless this journal has Results and Discussion grouped together, but I do not think it does.

Line 136: Not sure this is necessary to mention here

Line 178 (re: bottom-up control): Note that phytoplankton abundance may also be affected by top-down processes via iron contributions from top predators (whales or seabirds); Savoca and Nevitt 2014 in PNAS.

Para starting at line 242 (I mentioned this above): I found this paragraph to be only touching ever-so-briefly on a key discussion point, which is that silverfish are critical prey for lots of species in the Southern Ocean. To me, explaining this more and expanding on what changes in silverfish abundance and distribution could mean for predators would be important.

Line 307: The study region itself needs a map

Line 361: I would recommend having a separate map for the sampling grid and study area, and not combining it with a figure that shows the ASL examples, as Figure 1 does now.

Line 522: Can we see the AIC scores, residuals and deviance? It is curious to me that all your variables have p-values less than 0.05 - were there other variables you could have included? For example, what about lag of chlorophyll concentration? Ice concentration/extent? Bathymetry? (I don't know if ocean depth plays a role, but wondering if silverfish larvae are likely to be detected at a consistent depth regardless of depth of the ocean?) Also, I thought you included other variables like ENSO, etc.? Where are those results?

Reviewer #2 (Remarks to the Author):

The work by Corso et al. provides compelling evidence for the relationship between the Amundsen Sea Low (ASL) and Antarctic Silverfish larval abundance, using a well-designed GLM model integrating data from a 25-year time series. Based on established patterns of warming along the western Antarctic Peninsula (WAP), combined with the unique physiology of Antarctic fish and the life history of Antarctic Silverfish in particular, the authors predict that sea ice advance and sea surface temperature will be associated with larval abundance. They indeed find strong evidence for such a relationship, as depicted in Fig. 2a. The body of data available from the Palmer LTER time series and the strength of the model that the authors developed allows them to predict the impact of increased sea surface temperature (Fig. 2b), delayed sea ice advance (Fig. 2c), and autumnal ASL strength (Fig. 4) on larval abundance. The authors place these findings in the context global climate change and its predicted impact on the ASL, closing with the cascade of impacts that drops in Antarctic Silverfish abundance may incur on WAP food webs.

The authors do well to account for potential confounding factors in their model and conclusions:

- Their choice of model type in order to control for zeros in their dataset is well supported.
- They account for the significant correlation between larval abundance and chlorophyll concentration and surface salinity by discussing the influence of prey field dynamics.
- They provide well-referenced support for the main circulation patterns that influence climate in the WAP, including the increasing evidence for the importance of the ASL in particular. By including SAM and ENSO indices in their model development, the authors are able to tease apart the influence of ASL from these other climate modulators in the WAP.

My suggested revisions to the manuscript are included in the reviewer_track_changes manuscript file uploaded with this review. Importantly, the authors should take care in describing their conclusions – their results show evidence of significant correlation and provide compelling predictions of future tipping points, but do not support a causal relationship between environmental variables and larval abundance. Additional suggestions are largely editorial, with some comments particularly in the introduction suggesting edits to improve the presentation of information.

Overall I think the findings presented in this manuscript represent an important contribution to studies of the impact of climate change on biodiversity in Antarctica, highlighting important circulation patterns and environmental variables for researchers to focus on in future studies. In addition, the synthesis of the authors' findings into a model of Antarctic Silverfish reproduction in the WAP provides a valuable basis for future hypothesis testing in this species.

These review comments are also attached as a PDF to the review.

Reviewer #3 (Remarks to the Author):

General comments

The present paper aims at assessing and foreseeing the impact of climate change in the western Antarctic Peninsula (WAP) on the population dynamic of a key fish species of the Antarctic pelagic food web, such as the early life stages of the Antarctic silverfish (*Pleuragramma antarctica*). In particular, authors found a close relationship between the spawning success and larval abundance of this species and the strength and position of the Amundsen Sea Low (ASL), a climatological low-pressure center located in the Amundsen Sea, recently considered one of the most important factors involved in the sea ice loss and glacial retreat along the WAP. Based on a remarkable time series of data, this study is certainly novel, being the first attempt to clearly document the most important environmental drivers of the Antarctic silverfish population dynamics, as well as of interest for a wider audience. The statistical analysis of data is rigorous and appropriate, based on a robust sampling design. Findings are discussed in relation to current literature in a successful way, and results fully support conclusions.

Specific comments

Overall, I have no substantial criticism throughout the paper, except for what is shown in figure 1. In the case of weak ASL (fig. 1a), which represents a favourable situation for the forthcoming *Pleuragramma* larval abundance, its position is poleward respect to the case of strong ASL (Fig. 1b). It seems partially in contradiction with Fig. 4b (and on what is generally stated in the text), where the southward location of ASL would determine low larval abundance. Do strength and position of ASL always act synergistically, or they don't?

Mario La Mesa

14 September 2021

Dear Reviewers,

Thank you for your continued consideration of our manuscript. My coauthors and I appreciate the helpful comments we received from you all. We have responded to the Reviewers' specific comments below.

In addition to your comments, Reviewer #2 suggested several editorial changes to the manuscript through Track Changes in Word. We accepted each of these changes, which are highlighted in the revised Word file. Per the instructions of our editors, we also highlighted all other changes in the revised manuscript.

Please note, Reviewer #1 based their line-by-line comments on the Article File PDF, which apparently starts line numbering at the title automatically. However, Reviewer #2 added their in-text changes to our original Word file, which starts line numbering at the introduction section per the formatting request of *Nature*. As a result, the line-by-line suggestions of Reviewer #1 do not match up with the attached revised manuscript file. For clarity, we have also added Reviewer #1's comments in-text where they are meant to appear.

Thanks again,

Andrew D. Corso
PhD candidate, Fisheries Department, Virginia Institute of Marine Science (VIMS)

Deborah K. Steinberg
CSX Professor & Chair, Biological Sciences Department, VIMS

Sharon E. Stammerjohn
Senior Research Associate, Institute of Arctic and Alpine Research (INSTAAR), University of Colorado Boulder

Eric J. Hilton
Professor, Fisheries Department, VIMS

Reviewer #1

Remarks to the author:

- **Comment 1:** The overarching takeaway for me throughout was that there was a lack of central focus - for example, laying out specifically your hypotheses and/or objectives in the introduction would help.
 - **Response:** Objectives have been added to the last paragraph in the Introduction section (lines 55 to 59). The objectives also correspond to subheadings added in the Results and Discussion section to better lay out the components of our research.
- **Comment 2:** For example in the Results section, there is quite a bit of Discussion, in my view, which distracted me from your key findings. It was difficult to connect the results of your statistical models with your interpretation. I think a straight-forward presentation and keeping the sections separate and distinct from each other would have helped me comprehend the science a bit better.

Response: We understand the request to separate Results and Discussion to improve clarity, which was especially confusing given the Results section contained Discussion material. As we prefer to keep the sections combined, to facilitate immediate interpretation of the results, we checked with our editors (Drs. Grinham and Hoffmann) and *Communications Biology* does allow the combination of Results and Discussion if they are labeled as such. The *Communications Biology* style formatting guide also recommends information within Results and Discussion split into subheaded sections to benefit readability. We thus combined the headings into one 'Results and Discussion' section and added subheadings for our different research questions. As mentioned above, we believe the subheadings now help keep the reader focused on our objectives.

- **Comment 3:** I was also surprised to see a lack of discussion or focus on the implications of results for marine predators especially given the rebuttal letter to the editor, which noted the importance of penguin dynamics to Nature Climate Change readers as justification to reconsider this manuscript... I would agree with your logic, but then I would recommend making the connection to this point stronger in your manuscript. The fact that the ASL affects sea ice dynamics and wind, that that in turn affects abundance/distribution of larval silverfish is in itself interesting, but I think to round out the manuscript, paying more attention to this point you made (i.e., their importance in the food web as both predators and prey) would strengthen this justification for this manuscript to be considered for publication, as per your rebuttal letter.
 - **Response:** Thank you for this recommendation. Originally, we felt limited by space to further elaborate on food web impacts. However, with the relaxed length restrictions of *Communications Biology* we can and did expand our interpretations. We have made *Possible ecosystem responses to Antarctic Silverfish decline* a section subheading (line 220). In addition to the paragraph on

penguins, we added a paragraph on Antarctic toothfish/Weddell seals (lines 232 to 242) and another paragraph on zooplankton (lines 244 to 251).

- **Comment 4:** I would also like to note that silverfish are a key component in Weddell seal diet (far less so for crabeater seal, as in your rebuttal letter).
 - **Response:** Thank you, we corrected our assertion in the new paragraph (line 240).

Specific questions and feedback:

- **Comment 5,** Line 52: Here it might be worthwhile to note that in particular they can deposit eggs in frazil ice, which if I am not mistaken can be formed via strong, cold katabatic winds. If katabatic winds are less frequent, in addition to ice extent and ocean temperature (super-cooled waters for frazil ice formation), that could very well affect eventual larval abundance? Might be worth mentioning if this is correct, to strengthen your point.
 - **Response:** We agree this could strengthen our argument regarding the impact of winds, but found it difficult to discuss and define the concept in appropriate detail in the Introduction. Instead, we added this information on bottom ice types as nursery habitat to the Results and Discussion subheading *Connections with ASL strength and location*, on lines 151 to 161.
- **Comment 6,** Line 89: sampled? or detected? or both?
 - **Response:** We think “sampled” fits best in this paragraph (they were collected samples).
- **Comment 7,** Line 106-108 is Discussion, I think.
 - **Response:** Comment addressed above in response to Comment 2.
- **Comment 8,** Line 110-116 is probably better suited for Discussion. Typically, if something needs a reference to another study, you are inherently not talking about the results of your own study and in my view, it is not suitable for the results section - unless this journal has Results and Discussion grouped together, but I do not think it does.
 - **Response:** Comment addressed above in response to Comment 2.
- **Comment 9,** Line 136: Not sure this is necessary to mention here
 - **Response:** Since ASL longitude can also significantly impact WAP climate, we needed to mention somewhere in the manuscript why we did not consider it in our model. We moved it earlier in the paragraph where it fits/reads better.

- **Comment 10**, Line 178 (re: bottom-up control): Note that phytoplankton abundance may also be affected by top-down processes via iron contributions from top predators (whales or seabirds); Savoca and Nevitt 2014 in PNAS.
 - **Response:** We opted not to include the reference here as would be speculative to say the correlation we found involves sea birds, and in fact the tritrophic mutualism concept described in the paper would likely not lead to the positive relationship we found, i.e., more Chl a = more DMS released = more sea birds = less (not more) larval Silverfish as sea birds consume Silverfish (likely outweighing the advantage of Iron fertilization of phytoplankton by seabird excretion).

- **Comment 11**, Para starting at line 242 (I mentioned this above): I found this paragraph to be only touching ever-so-briefly on a key discussion point, which is that silverfish are critical prey for lots of species in the Southern Ocean. To me, explaining this more and expanding on what changes in silverfish abundance and distribution could mean for predators would be important.
 - **Response:** Comment addressed above in response to Comment 3 (and presume reviewer meant line 224).

- **Comment 12**, Line 307: The study region itself needs a map
 - **Response:** Agree, we added a map dedicated to study region (Supplementary Fig. 1). To illustrate the seasonality of regional sea ice, we also included summer (DJF) and winter (JJA) sea ice concentrations on the map.

- **Comment 13**, Line 361: I would recommend having a separate map for the sampling grid and study area, and not combining it with a figure that shows the ASL examples, as Figure 1 does now.
 - **Response:** Comment addressed above in response to Comment 2.

- **Comment 14**, Line 522: Can we see the AIC scores, residuals and deviance? It is curious to me that all your variables have p-values less than 0.05 - were there other variables you could have included? For example, what about lag of chlorophyll concentration? Ice concentration/extent? Bathymetry? (I don't know if ocean depth plays a role, but wondering if silverfish larvae are likely to be detected at a consistent depth regardless of depth of the ocean?) Also, I thought you included other variables like ENSO, etc.? Where are those results?
 - **Response:** We added a table showing the AIC/ Δ AIC/deviance of the different temporal lags (Supplementary Table 1), another table showing the comparative impact of ASL/SAM/ENSO on the AIC/ Δ AIC/deviance (Supplementary Table 2), and two diagnostic residual plots for the final model calculated using the

DHARMA package (Supplementary Figure 5). The QQ plot shows some slight underdispersion, but for such a large time series we are pleased with the residual trends in the final model.

Regarding p -values, that table (now Supplementary Table 3) shows the results of the conditional component from the final model. Indeed, these parameters were statistically significant to varying degrees, but the final model was selected using AIC, parsimony, and hypothesis testing (i.e., we did not ‘ p -mine’).

We did consider several other variables that were not selected in the final model. For example, in the Model development section of Materials and Methods (lines 388 to 405), we mention that temperature/salinity at various depths (surface to 120m), sea ice duration, extent, day of retreat, day of advance, ENSO, SAM, and ASL were evaluated, including all four seasons for the ASL RCP and latitudinal location (DJF, MAM, JJA, & SON). While a table that includes all model combinations of environmental variables that we considered could be added, we decided the length and jargon would take away from the conclusions of the paper. Instead, we added a simplified table that shows the impact of ASL, SAM, and ENSO on model performance (Supplementary Table 2) and a table that shows the impact of different temporal lags on these variables (Supplementary Table 1).

Although we did not mention ocean bathymetry in Materials and Methods, we did evaluate it as a model parameter. However, it did not improve model performance and caused convergence problems when included. We amended Materials and Methods to include bathymetry (lines 393 to 394). In the Model development section we state that LTER sampling stations (i.e., net tow coordinates) were included as a random variable in the final model to account for unobserved spatial heterogeneity (lines 424 to 425). As the tow coordinates are correlated with bathymetry (e.g., stations with a lower value are closest to the coast and shallowest), we are satisfied that depth variability was sufficiently incorporated into the model for the purposes of this analysis.

Reviewer #2

- **Comment 1:** Given the correlational nature of the study, I think it would be best not to use a causal verb like ‘impact’ here, but rather something along the lines of ‘are associated with reduced’ instead of ‘negatively impact’.
 - **Response:** We changed the wording as suggested.
- **Comment 2:** Alternatively, ‘, and’
 - **Response:** Left reviewer’s semicolon suggestion instead.

- **Comment 3:** I would rephrase this because there is actually strong continental shelf circulation in the WAP, but it is due to the ACC, not the Slope Current. And since the ACC does not approach the shelf in the Ross and Weddell Seas, it is a less effective circumpolar transport mechanism than the Slope Current. The ACC may be involved in shelf transport of individuals along the peninsula, but beyond that, its role is less clear. Your point regarding the absence of the Slope Current reducing connectivity between WAP populations and others remains valid – I would simply remove the highlighted text about weak shelf circulation.

- **Response:** Agreed, removed as suggested.

- **Comment 4:** Is 5°C really ‘just’? This seems like a large temperature increase, given your following sentence that water temperatures only increased by 1°C over a 75-year period (which is still a lot, but not compared to 5°C!). I would rephrase to either remove 5°C and just say ‘increased temperatures’, or rephrase to remove the modifier ‘just’.

Alternatively, if the ‘just’ is here because non-polar species can adjust to much larger temperature ranges, I may add a sentence about that to provide a context as to why these effects are so significant, given that non-polar species are not impacted at this level of temperature increase.

- **Response:** We have removed the modifier.

- **Comment 5:** I would change this to ‘reproduce’, as the relevance of sea ice is not just that the eggs are deposited there, but also because the adults return there when they reproduce, and also because the early larval stages feed on plankton endemic to the platelet ice layer below sea ice, which also provides them with a physical protection against predators.

- **Response:** We agree that more information is necessary to convey the importance of sea ice as habitat throughout the life history of Antarctic Silverfish. See new edits on lines 23 to 24.

- **Comment 6:** References 13 (Kellerman 1996), 14 (La Mesa 2015), 15 (Parker 2015), and 18 (Ross 2014) go well here, as they all deal specifically with the WAP, but I would remove reference to 16 (Koubbi 2011), 17 (Davis 2017), and 19 (La Mesa 2010), as they are all focused on silverfish in the Ross Sea and make little mention of silverfish abundance in the WAP.

- **Response:** Agreed. In order to give Koubbi, Davis, and La Mesa credit for their contributions to our hypotheses, we now reference these papers in context of ‘other coastal regions of the Southern Ocean’. See new edit on lines 28 to 29.

- **Comment 7:** As in the abstract, this sentence presents a causal relationship that these data cannot support. An alternative phrasing could be: ‘The significant relationship presented here between sea ice and ocean temperatures and larval abundance along the

WAP supports the hypothesis that the ASL impacts Antarctic Silverfish spawning behavior’.

- **Response:** Agreed, thank you for the alternative phrasing, which we slightly further modified; new text is on lines 201 to 203.
- **Comments 8, 9, and 10:** Standardize referral to Materials and Methods as ‘Materials and Methods’.
 - **Response:** Done.
- **Comment 11:** Reference needs to be converted to correct style
 - **Response:** Changed.
- **Comment 12:** This link did not work for me when I tried to open it – perhaps it needs to be updated?
 - **Response:** A new, working link has been added. (<https://psl.noaa.gov/enso/mei/>)
- **Comment 13:** Reference needs to be converted to correct style
 - **Response:** Changed.

Reviewer #3

- **Comment 1:** Overall, I have no substantial criticism throughout the paper, except for what is shown in figure 1. In the case of weak ASL (fig. 1a), which represents a favourable situation for the forthcoming Pleuragramma larval abundance, its position is poleward respect to the case of strong ASL (Fig. 1b). It seems partially in contradiction with Fig. 4b (and on what is generally stated in the text), where the southward location of ASL would determine low larval abundance. Do strength and position of ASL always act synergistically, or they don’t?
 - **Response:** The short answer is no, strength and latitudinal position of the ASL do not act synergistically during austral autumn (although strength and longitudinal position do appear to be correlated during parts of the year).

However, as the reviewer mentions, this does allow for the possibility of contradicting scenarios for Antarctic Silverfish. An anomalously strong, or deep, ASL RCP during MAM (poor conditions for larvae) could be paired with a more northward central location (ideal conditions for larvae). An obvious follow-up question is: which component of the ASL (strength or location) is more important for controlling larval abundance? The model estimate for ASL RCP is slightly larger than latitude (Supplementary Table 3), which gives some indication that RCP may account for more of the variability in larval abundance. However, an additional future study that considers ocean circulation and winds is necessary to accurately address this question.

We mention in lines 144 to 146 that we did not observe collinearity between RCP and latitude. However, to prevent confusion for readers we also added a sentence to Figure 4 (lines 347 to 348) that reinforces the lack of relationship between RCP and latitude.

Reviewers' comments:

Reviewer #1 (Remarks to the Author):

The authors did a nice job of implementing feedback from previous reviews and I think this is an interesting paper. My feedback and questions are in the attached PDF file, but largely speaking I think the authors should consider being careful with the language being used (e.g., significant, impact, likely) relative to the results. It was not clear to me the strength of the influence of each variable within the models (the p-value is only one piece of the story) and I'm concerned the argument is suggesting a clearer influence of ASL than is warranted based on the data.

Looking at the CPUE data, I'm also wondering why a log transformation wasn't done? I'm less familiar with zero-inflated models, so maybe that takes care of it, but the graphs showing 0.002 larvae per 1000 cubic meters makes me question whether all those zeroes were actually accounted for? Just seems like a transformation to something that makes a little more sense would be useful.

Reviewer #2 (Remarks to the Author):

The authors have done an excellent job in responding to the comments of the reviewers, particularly reviewer 1, who made some excellent points regarding the presentation of the results and discussion, as well as asking relevant questions regarding the model criteria. Between the authors' extensive edits of the manuscript, the addition of further supplementary figures and tables, and the merging of the results and discussion into a series of focused sub-sections, I think that the manuscript has greatly improved, and is ready for publication. I have no further comments, and laud the authors for their efforts in revision, and for carrying out this important work.

General comments:

Comment 15: It was not clear to me the strength of the influence of each variable within the models (the p-value is only one piece of the story) I'm concerned the argument is suggesting a clearer influence of ASL than is warranted based on the data.

Response: See response to Comment 26. Additionally, we present model performances with the impacts of the ASL, SAM, and ENSO respectively in Supplementary Table 3. This table clearly demonstrates the superior model performance when considering the sole impact of the ASL compared to these other two influential climactic features.

We do not argue a “clear” influence of the ASL. In fact, on lines 144 – 146 we state, “additional ocean modeling is required to verify specific causes of the correlation between ASL strength, location, and adult fish”. However, we do argue that the ASL plays an important role in shaping this regional population of Antarctic Silverfish, and we are comfortable with how this argument is presented in our manuscript.

Comment 16: Looking at the CPUE data, I'm also wondering why a log transformation wasn't done? I'm less familiar with zero-inflated models, so maybe that takes care of it, but the graphs showing 0.002 larvae per 1000 cubic meters makes me question whether all those zeroes were actually accounted for? Just seems like a transformation to something that makes a little more sense would be useful.

Response: In *Nature Proceedings*, O'Hara and Kotze (2010) write, “Ecological count data (e.g., number of individuals or species) are often log-transformed to satisfy parametric test assumptions. Apart from the fact that generalized linear models are better suited in dealing with count data, a log-transformation of counts has the additional quandary in how to deal with zero observations. With just one zero observation (if this observation represents a sampling unit), the whole dataset needs to be fudged by adding a value (usually 1) before transformation. Simulating data from a negative binomial distribution, we compared the outcome of fitting models that were transformed in various ways (log, square-root) with results from fitting models using Poisson and negative binomial models to untransformed count data. We found that the transformations performed poorly, except when the dispersion was small and the mean counts were large. The Poisson and negative binomial models consistently performed well, with little bias.”

We emphasize the problematic mathematical issue of log transforming a value of zero. Martin et al. 2005 write, “while the transformation may normalize the distribution of the non-zero values, no transformation will spread out the zero values. The high frequency of zero values is simply replaced by an equally high frequency of the value to which zero is transformed”.

Although we used density (based on counts) rather than raw count data in our analyses for this manuscript, the same weaknesses described above apply. With modern computing power, it is no longer necessary to log transform data to fit normal (Gaussian)

distributions (Steel et al. 2013). Although our model choice (GLMM) has its own downfalls (Ives 2015), we are confident about its superior performance in this situation. We developed our zero-inflated model and selected our distribution (Tweedie) to fit our data without unnecessary transformations (see Supplementary Figure 5).

Regarding the question of interpretability, the predicted impacts (i.e., estimated marginal means) in our analysis (Figures. 2c, 2b, 4a, 4b and Supplementary Figures 3a and 3b) are meant to serve as indicators of relative abundance. The small values of larvae (e.g., 0.01/1000m³) on the y-axes are indicative of a non-targeted sampling design incidentally capturing larvae with a pre-existing patchy distribution. We could alter the values to be more visually appealing, but we prefer to present the data as they are. There are several recent studies in ecological literature which use marginal means calculated from zero-inflated models with similarly small scales. For example, see Ivory et al. 2019 (they show mean densities of 0.0007 larval eels per m³), or Wismer et al. 2019, *Communications Biology* (they show damselfish densities as low as 0.1 individuals/m³).

References:

- O'Hara, R. and Kotze, J., 2010. Do not log-transform count data. *Nature Precedings*, pp.1-1.
- Martin, T.G., Wintle, B.A., Rhodes, J.R., Kuhnert, P.M., Field, S.A., Low- Choy, S.J., Tyre, A.J. and Possingham, H.P., 2005. Zero tolerance ecology: improving ecological inference by modelling the source of zero observations. *Ecology letters*, 8(11), pp.1235-1246.
- Steel, E.A., Kennedy, M.C., Cunningham, P.G. and Stanovick, J.S., 2013. Applied statistics in ecology: common pitfalls and simple solutions. *Ecosphere*, 4(9), pp.1-13.
- Ives, A.R., 2015. For testing the significance of regression coefficients, go ahead and log-transform count data. *Methods in Ecology and Evolution*, 6(7), pp.828-835.
- Ivory, J.A., Steinberg, D.K. and Latour, R.J., 2019. Diel, seasonal, and interannual patterns in mesozooplankton abundance in the Sargasso Sea. *ICES Journal of Marine Science*, 76(1), pp.217-231.
- Wismer, S., Tebbett, S.B., Streit, R.P. and Bellwood, D.R., 2019. Young fishes persist despite coral loss on the Great Barrier Reef. *Communications biology*, 2(1), pp.1-7.

Line by line comments:

Comment 17, Line 43: Of the ocean, right? What RCP scenario does this happen? Might be good to put this in context, how likely/possible the 5 degrees of warming of the ocean is.

Response: The temperature increase we cited here was conducted in an experimental setting and is not related to an RCP scenario; we now clarify this in the text. We also provide additional context for the observed ocean warming in regards to heat intolerance.

Comment 18, Line 86: Not sure I saw the prediction analysis?

Response: See lines 439 – 445 in Materials and Methods. Predictions were calculated using the `ggpredict` function in the `ggeffects` package. To prevent confusion, we have added, “Based on estimated marginal means,” the first time we use the word “predict” on lines 73 – 74. See the vignette below to further understand the definition of the term “prediction” in relation to estimated marginal means.

https://strengejacke.github.io/ggeffects/articles/introduction_marginal_effects.html

Specifically, Daniel Lüdecke writes, “In Stata language, they are also called “predictive margins”, and this is what `ggeffects` returns. Thus, the language used throughout this package considers “marginal effects” as predictions, i.e. predicted values. Depending on the response scale, these are either predicted (mean) values, predicted probabilities, predicted (mean) count (for count models) etc.”

Comment 19, Line 97: Where were they mostly caught, though? Was this equally across all sampling locations? Or were they more likely to be caught at certain stations?

Response: We observed no significant spatial autocorrelation between stations, see Materials and Methods (lines 427 – 429).

Comment 20, Line 104: Wait... so they need the frazil ice but are found at 120 meters? So if you didn't find them at 120 meters when the water was warmer, couldn't they have just moved deeper given they are neutrally buoyant?

Response: As fish larvae develop generally, their habitat preferences also change. Current literature suggests Antarctic Silverfish larvae prefer the protective, nutrient-rich environment of sea ice (especially frazil ice) during their early development (i.e., eggs and preflexion). As the larvae grow and become stronger swimmers (postflexion), they eventually move to other pelagic habitats.

As specified in Materials and Methods (lines 365 – 366), the net is towed to 120m and then retrieved. These are not discrete vertical tows. Therefore, we cannot speak to the vertical movements of Antarctic Silverfish larvae in this area. This is one reason why we are using a zero-inflated model in our analysis. In Materials and Methods (lines 419 – 422), we write, “...there is significant overrepresentation of “zero fishes captured” each year at stations along the Palmer LTER sampling grid. Zero-inflated GLMMs were selected over zero-altered, as any zeros in the larval Antarctic Silverfish time series are likely ‘false’, resulting from an imperfect sampling design.”

Comment 21, Line 113: What is the magnitude of the effect?

Response: See Figure 2c to visually interpret the magnitude of this variable.

Comment 22, Line 132: Hmm, I would be careful about the word prediction here. I think you're using the results of your explanatory model to suggest this, no? That's not really predicting anything.

Response: See response to Comment 18.

Comment 23, Line 137: These models are correlative only, I would highly recommend being careful with your language to be clear about that, throughout.

Response: We are comfortable with this language, and this phrasing seems to have been acceptable to the other reviewers.

Comment 24, Line 142: Ok p-value is good but what's the strength of the relationship compared to the other variables? Just because a variable has a low p-value doesn't mean it has a strong impact on the overall model.

Response: Agreed, see Supplementary Tables 1 to find the estimates for each of our fixed-effect variables, supplementary Table 2 to see the impacts of 0, 1, and 2-year temporal lags on model performance, and Supplementary Table 3 to view the impacts of ASL, SAM, and ENSO on model performance.

Comment 25, Line 152: larvae abundance?

Response: Changed.

Comment 26, Line 153: If you can add the detail about the magnitude/strength of the association in text that would be really helpful (coefficients and z-scores).

Response: We added references to Supplementary Table 1, where the reader can find coefficients and Wald z score for each of the fixed-effects variables in our model.

Comment 27, Line 201: Same comments as above re: the magnitude of the associations here.

Response: See Supplementary Table 1 to find the estimates for each of our fixed-effect variables.

Comment 28, Line 211: It would be nice to see where these locations are on the map of points.

Response: Salinity was not the focus of this manuscript (which is why the figures are Supplementary material). It would take a prohibitive amount of analytical effort to create the map, given it would not be critical to our study.

Comment 29, Line 239: Wait did you say you checked for collinearity among ASL and all these variables?

Response: Yes we did, see lines 148 – 151, 352 – 353, and 410 – 413.

Comment 30, Line 253: "possibly" instead of "likely"?

Response: Changed.

Comment 31, Line 260: In a previous review I recommended adding detail regarding predators but I should be clear that while you do that, you should stay away from conjecture. I'm pretty sure the number of toothfish taken out of the Amundsen and Bellingshausen Seas is next to

nothing. I would therefore recommend carefully considering if you want to include this paragraph.

Response: It is correct that there is not currently a major *fishery* in the Amundsen and Bellingshausen Seas for Antarctic Toothfish. However, despite the lack of fishing effort, these areas still represent important habitat for Antarctic Toothfish.

The Amundsen and Bellingshausen Seas are split into two, data-limited Subareas: 48.1 and 88.3. Currently, fishing is prohibited in both areas. Although, CCAMLR found Subareas 48.1 contained a CPUE (g/hook) of 18.2 for *Dissostichus mawsoni*. In only 24 hauls, CCAMLR scientists caught 890 kg of Antarctic Toothfish during an exploratory study. There is also an ongoing exploratory fishing effort in Subregion 88.3. In 2020, just two vessels caught 96 tons of Antarctic Toothfish.

Clearly, there is a poorly understood population of Antarctic Toothfish in these areas. We also checked the validity of this paragraph (i.e., if there was too much conjecture) with a current member of the CCAMLR Scientific Committee (Dr. Christopher Jones). The paragraph accurately represents current knowledge in the field without suggesting erroneous implications of our results.

References:

http://archive.ccamlr.org/ccamlr_science/Vol-06-1999/01arana-vega.PDF
https://fishdocs.ccamlr.org/FishRep_883_TOA_2020.pdf

Comment 32, Line 261: Are these prevalent in these regions?? I think it's elegendoides (mostly) outside of the Ross Sea, so just be careful and clear about what you're saying.

Response: See response to Comment 31.

Comment 33, Line 284: How does this study compare (regarding abundances and locations, etc.) with other regions? I think you're missing some key papers, for example, the one by Brooks et al. 2018 (Early life history connectivity of Antarctic silverfish (*Pleuragramma antarctica*) in the Ross Sea).

I could be confused but it seems like the abundances per sample of water were always way lower than what was caught in the Ross Sea... what does that mean for your study? I think adding some context would be good.

Response: It is not possible to directly compare the abundances found by Brooks et al. 2018 with our results. In their study, efforts to capture Antarctic Silverfish larvae were *targeted* in inflows and outflows that are known to transport larvae from their regional nursery ground – Terra Nova Bay. In our study, Antarctic Silverfish larvae were captured *incidentally* by tows designed to collect zooplankton, such as krill and salps. Due to differences in sampling design, we refrain from comparing our data to other regions in this initial analysis. With the proper assumptions and analytical methods, a separate review of the current knowledge of Antarctic Silverfish larvae in Southern Ocean would be a more appropriate setting for this complex comparison.

Comment 34, Line 361: So, over the course of 25 years you caught 7500 fish or so... again maybe I'm just misunderstanding but having an increase of 0.001 fish (I don't even know how to interpret that) per approximately 4-6 degrees of latitude... what does that even mean??

Do you have to sample 1 million cubic meters of water before you get a single larvae? This is accounting for zero inflation?

Response: See response to Comment 16.

Comment 35, Line 386: grid based on what though?

Response: We added Smith et al. 1995 and Ducklow et al. 2007 as references to this line (364) in our manuscript for readers that are further interested in the creation of the grid and science of the Palmer LTER.

The Palmer LTER research grid was initially established to: (1) document the interannual variability of the cryosphere, (2) identify forces that cause variation in physical forcing, (3) construct models that link ecosystem processes to environmental variables, and (4) employ such models to predict and validate ice-ecosystem dynamics. See Smith et al. 1995 for more information.

References:

Smith, R.C., Baker, K.S., Fraser, W.R., Hofmann, E.E., Karl, D.M., Klinck, J.M., Quetin, L.B., Prézelin, B.B., Ross, R.M., Trivelpiece, W.Z. and Vernet, M., 1995. The Palmer LTER: A long-term ecological research program at Palmer Station, Antarctica. *Oceanography*, 8(3), pp.77-86.

Ducklow, H.W., Baker, K., Martinson, D.G., Quetin, L.B., Ross, R.M., Smith, R.C., Stammerjohn, S.E., Vernet, M. and Fraser, W., 2007. Marine pelagic ecosystems: the west Antarctic Peninsula. *Philosophical Transactions of the Royal Society B: Biological Sciences*, 362(1477), pp.67-94.

Comment 36, Line 421: Please clarify how this was done, I'm confused about the use of the stations as proxies for bathymetry. If you have bathymetry data you should use it rather than station numbers.

Response: As we mentioned in our last response (to Comment 14), bathymetry data was used in our analysis and caused convergence issues with our model.

Net tows take place at sampling stations with fixed locations on the Palmer LTER research grid. Each station has an associated depth that does not change. As we state on lines 429 – 430, we included the coordinates (in units of UTMs) of each net tow as a random effect in our model. *If* larval abundance was impacted by bathymetry (which we do not believe to be the case due to the lack of spatial autocorrelation), then a sufficient amount of that variability would be captured by the tow coordinates as a random effect. For example, if larvae were more frequently caught at shallower neritic depths, then those near-shore stations (i.e., those with shallower depths) would routinely have higher abundances of larvae, and that correlation would be modeled as a random effect.